# A Novel Approach for Emotion Recognition Based on EEG Signal Using Deep Learning

Awf Abdulrahman [1], Muhammet Baykara [2] and Talha Burak Alakus [3],*

1 Department of Public Health, Duhok Polytechnic University, Duhok 42001, Iraq
2 Department of Software Engineering, Fırat University, Elazığ 23119, Turkey
3 Department of Software Engineering, Kırklareli University, Kırklareli 39000, Turkey
* Correspondence: talhaburakalakus@klu.edu.tr

**Abstract:** Emotion can be defined as a voluntary or involuntary reaction to external factors. People express their emotions through actions, such as words, sounds, facial expressions, and body language. However, emotions expressed in such actions are sometimes manipulated by people and real feelings cannot be conveyed clearly. Therefore, understanding and analyzing emotions is essential. Recently, emotion analysis studies based on EEG signals appear to be in the foreground, due to the more reliable data collected. In this study, emotion analysis based on EEG signals was performed and a deep learning model was proposed. The study consists of four stages. In the first stage, EEG data were obtained from the GAMEEMO dataset. In the second stage, EEG signals were transformed with both VMD (variation mode decomposition) and EMD (empirical mode decomposition), and a total of 14 (nine from EMD, five from VMD) IMFs were obtained from each signal. In the third stage, statistical features were obtained from IMFs and maximum value, minimum value, and average values were used for this. In the last stage, both binary-class and multi-class classifications were made. The proposed deep learning model is compared with kNN (k nearest neighbor), SVM (support vector machines), and RF (random forest). At the end of the study, an accuracy of 70.89% in binary-class classification and 90.33% in multi-class classification was obtained with the proposed deep learning method.

**Keywords:** EEG; emotion recognition; deep learning; signal processing

## 1. Introduction

Emotion is defined as the reaction or consciousness to external stimuli. It plays an important role in daily life as it affects people's routines. Basic emotions, such as happiness, anger, and sadness, are constantly reflected by individuals, voluntarily or involuntarily, and this significantly affects the position of individuals in society. People with negative emotions can be excluded by society and affected by this situation both physiologically and psychologically [1]. In addition, people with positive emotions have better living standards and live longer [2]. There are many studies on emotion analysis in the literature. Recently, researchers have been actively working in this field to understand and analyze the behavior of emotions. However, the fact that the emotions are abstract and vary from person to person causes the studies to be inefficient and the studies in this field do not reach the desired level [3]. In addition, a large number of data collection (facial expressions, voice signals, brain signals, body language, etc.) and analysis methods cause the data to be complex and the analysis process to take a long time. For these reasons, the need to use computer-aided and artificial intelligence-based analysis methods has arisen [4].

Emotions can be obtained by various methods including voice signals, facial expressions, body language, and physiological signals. People convey their emotions through voices, words, and behavior. Emotion prediction is performed by analyzing the intensity of the voice, the strength of the voice, the speech level, and the speaking rate [5]. Another

method used to obtain emotions is the analysis of facial expressions. In this method, the subjects stand in front of the camera and their reactions to the stimuli are observed [6]. Similarly, emotions are examined with body language and physical movements. For example, while an angry person joins his/her lips, a bored person looks out of the window or bangs his/her feet [7]. However, the constant manipulation of the emotions obtained by these methods may cause the data to be inaccurate and unreliable. People can easily manipulate their gestures, tones, and movements [8]. For these reasons, the need for a more reliable system has arisen and the importance of physiological signals has increased. The most commonly used signals in emotion recognition studies are EEG (electroencephalography), ECG (electrocardiography), PPG (photoplethysmography), GSR (galvanic skin response), and SKT (skin temperature) [9]. However, the most preferred one among these methods is the EEG signals. The main reasons for this situation are that it is easy to use, cost-efficient compared to other methods, and has portable and wearable technology [10]. EEG is a technique for collecting data from a person by placing a wearable device around the head in a non-invasive way. This device contains electrodes placed around the head according to the 10–20 system, as given in Figure 1.

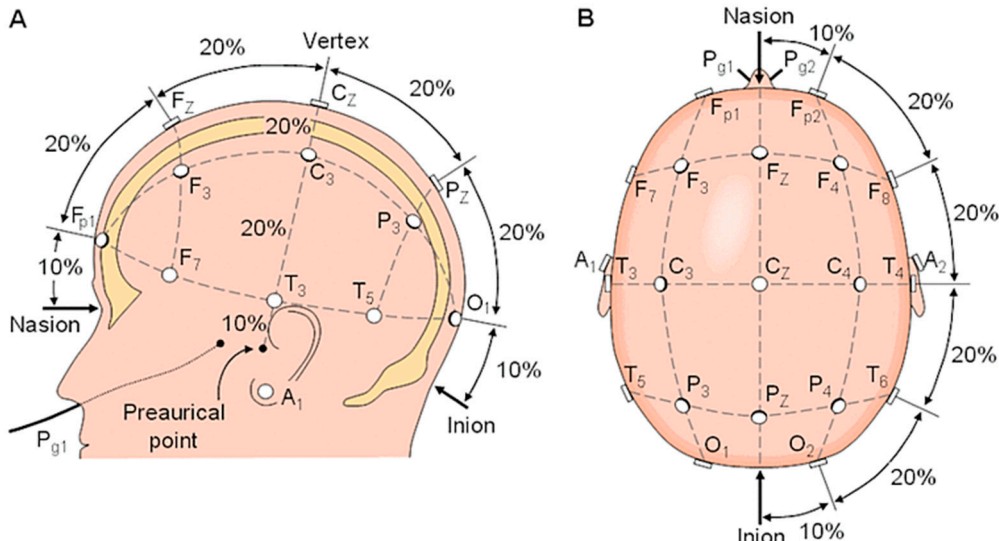

**Figure 1.** The 10–20 system. (**A**) front side, (**B**) top side [11,12].

This device monitors brain activity and records signals from the brain. These devices transmit the recorded signals to computers via wired or wireless communications for the purpose of analysis. EEG is an easy and inexpensive method that has been used to record brain activity, and it has also proven that it is through brain activity that emotions can be detected. Measuring brain activity via EEG devices helped scientists understand the emotions of people with disabilities whose emotions are difficult to detect through their facial expressions [13].

There are two types of emotion models in the literature, discrete and dimensional. The discrete emotion model consists of eight (anger, fear, sadness, disgust, surprise, anticipation, trust, and joy) basic emotions, including positive and negative emotions [14]. On the contrary, in the dimensional model, emotions are expressed not by their names but by their position in the arousal-valence plane [15]. Emotions are placed in four different zones on the arousal-valence plane. The valence plane indicates negative and positive emotions. The right side of the plane specifies positive valence, while the left side indicates negative valence. On the arousal plane, emotions are sorted from inactive emotions to active emotions from bottom to top. In Figure 2, a general structure of the dimensional emotion model is given.

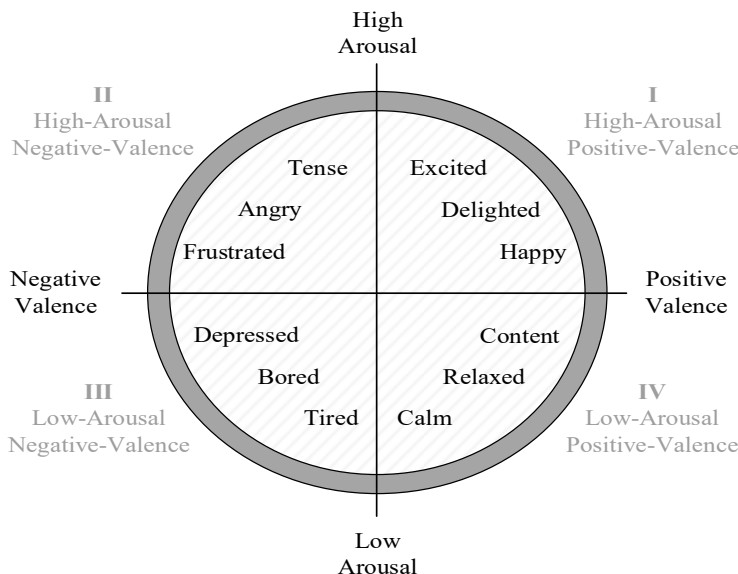

**Figure 2.** General structure of arousal-valence dimension.

As can be understood from Figure 2, the arousal-valence plane is divided into four separate zones. While HAPV (High Arousal Positive Valence) emotions are placed in the first zone, there are HANV (High Arousal Negative Valance) emotions in the second zone. On the contrary, there are low arousal feelings in the third and fourth zones. While the third zone consists of LANV (Low Arousal Negative Valance) emotions, the fourth zone consists of LAPV (Low Arousal Positive Valence) emotions. Thanks to this model structure, emotions are expressed not by their names but by their position in the plane. Therefore, this model is expressed as a universal model [16]. For example, the feeling of "bored" is now expressed as LANV.

Further, in the literature, there are three different types of stimuli (aural, visual, aural/visual stimuli), where stimuli are used to stimulate a person to reach the emotions to be extracted [17]. Aural stimuli are made up of sounds. It usually consists of music, sound clips. Visual stimuli are like pictures or images labeled with different names. As for aural/visual stimuli, which are considered among the best stimuli for extracting emotions, according to most studies [18,19], include both sounds and visuals.

In this study, the emotion recognition process was performed using both the dimensional emotion model and the discrete emotion model. The study consists of four stages. In the first stage, EEG emotion data were obtained from the GAMEEMO dataset. In the second stage, EEG signals were decomposed by EMD (Empirical Mode Decomposition) and VMD (Variational Mode Decomposition) and a total of 14 IMF (Intrinsic Mode Functions) were obtained for each EEG signal. Nine of the IMFs were obtained from EMD and the remaining five from VMD. Then, statistical features were extracted from IMFs and maximum value, minimum value, and average values were taken into consideration for the feature extraction process. In the last stage of the study, the classification was carried out with the designed DeepBiLSTM (Bidirectional Long-Short Term Memory) method and the performance of the classifier was evaluated with the accuracy value. The performance of the DeepBiLSTM method was also compared with kNN (K-Nearest Neighbors), SVM (Support Vector Machines), and RF (Random Forest). While positive and negative emotions were used for the binary-class classification process, the dimensional model was used for the multi-class classification process. In the binary-class classification process, the proposed deep learning model achieved an average accuracy of 70.89%, this rate increased in multi-class classification and was 90.33% on average. The main contributions of the study can be listed as follows:

- To the best of our knowledge, for the first time in this study, data belonging to the GAMEEMO dataset was analyzed with the DeepBiLSTM model.
- With this study, it has been observed that the deep learning model achieves better performance than machine learning models.
- EEG signals belonging to the GAMEEMO dataset were obtained with a portable EEG device. With this study, it has been observed that portable EEG devices are at least as successful as conventional devices.
- It was observed that the discrete emotion model and the dimensional emotion model performed differently.

The reason for choosing the GAMEEMO dataset is that it is a physiological dataset based on aural/visual stimuli (computer games). The aim of this study is to process biomedical signals to discover the effect of computer games on the players' personal mood (emotion). Moreover, this dataset is compatible with the discrete emotions model and the arousal-valence model (dimensional model). The rest of the work is designed as follows: In Section 2, emotion analysis studies in the literature are mentioned. The datasets used in these studies, suggested methods, and performances of classifiers were examined. In the Section 3, technical information about the dataset used in the study is given. In addition, in this section, general information about the feature extraction methods and classifiers is given. In Section 4, application results are given, and the performances of the classifiers are compared. In addition to these, the advantages and disadvantages of the study are given and discussed. In Section 5, the study is concluded and information about future studies is given.

## 2. Related Works

In this section, recent emotion recognition studies with EEG signals are included. In the study of [19], a new emotion dataset based on physiological brain signals was presented, called GAMEEMO. Four computer games (boring, quiet, horror, and funny) were used as visual/aural stimuli. Then, data were collected from 28 subjects using a device called EMOTIV EPOC with 14 channels. After collecting the data, they processed the collected data and the features were extracted by using the statistical method, DWT (discrete wavelet transformation), Hjorth, and others. Finally, MLPNN (multi-layer perceptron neural network), SVM, and kNN classifiers were applied to classify the discrete emotions and the arousal-valence model. In the study of [20], a discrete emotion model was used to classify the GAMEEMO dataset as a physiological database, and the features were extracted using the method of spectral entropy values of EEG signals, and then a bidirectional long-short term memory algorithm was used to classify the emotion. This study had 76.91% accuracy and 0.9 ROC (receiver operating characteristics). In the study of [21], a model based on an EEG was presented to detect real emotions, realistic emotions, and fake emotions. A model was created for rating real, fake, and neutral smiles. Features were extracted using three methods: DWT, EMD, and incorporating DWT into EMD. Then, kNN, SVM, and ANN (artificial neural network) were applied to classify the three emotions. The highest classification accuracy obtained from ANN, in the alpha and beta bands (94.3% and 84.1%, respectively). In the study of [22], an approach based on VMD was proposed to extract features from EEG signals and DNN (Deep Neural Network) was applied as a classifier. A dimensional emotion model was considered on the DEAP dataset. This study had a rating accuracy of 62.50% in valence and 61.25% in arousal. In the study of [23], the authors proposed a method for detecting personality traits by using physiological signals. Where the research team displayed a set of image and video stimuli on 18 subjects, data were collected from the participants in the experiment using a portable and wearable EEG device. The research team used the one-way ANOVA (analysis of variance) method to select the lowest set of features, then machine learning algorithms were applied including kNN, LR (logistic regression), NB (naïve Bayes), and SVM. The research team obtained the following accuracy results: 89.24% with kNN, 61.11% with LR, 79.17% with NB, and 95.45% with SVM. Authors in [24] performed a binary classification to detect stress using brain signals. They

collected data from the physical bank which consisted of 182 s of normal relaxation and 62 s of stress statuses. They calculated PSD (power spectral density) from the preprocessed data to extract the features using an FFT (fast Fourier transform). After extracting the features, they applied SVM and kNN algorithms to provide a classification. They obtained an accuracy of 99.42% from the FP1 EEG channel by using kNN, and 98.72% from the FP1 EEG channel with SVM. In the study of [25], the authors performed emotion recognition by using facial expressions. In the study, researchers obtained the emotion stimuli data from GAPED dataset. Researchers played music clips to stimulate the emotions of the participants in the experiment to discover the six basic emotions (joy, surprise, sadness, fear, disgust, anger) and the neutral state. The authors employed three types of SVM classifiers, including linear, RBF (radial basis function), and polynomial. At the end of the study, they achieved the best classification accuracy of 58% with a linear kernel. In the study of [26], the authors developed a new method for extracting emotions by combining HOG (oriented gradient) with GC (Granger causal), and TE (transfer entropy). SVM algorithm was applied to classify the emotional states of stress and calm. They obtained EEG data from DEAP dataset in their experiment where the data was reduced to improve processing and analysis time to 10 EEG channels. They obtained an average accuracy of 88.93% and 95.21% for both GC and TE with HOG, respectively. Authors in [27] proposed a novel multimodal framework for extracting human emotions using the DEAP dataset. Initially, the research team used three approaches to identify human emotions. The first one was the emotion recognition approach with EEG signals, and the second one involved motion recognition via face data. The last way to recognize emotions involved a fusion-based approach. In these approaches 3D-CNN (convolutional neural network), Mask-RCNN, and SVM methods were employed. The best accuracy was achieved from the 3D-CNN classifier with 96.13% and 96.79% for the equivalence and excitation categories. Authors in [28] proposed a new system to identify the negative emotions that may result from negative news on social media, using EEG signals. For the experiment, 10 participants were used in the study. Then, data were collected with an eight-channel EEG device. To obtain negative emotions, subjects examined 30 articles from different sources. After applying feature extracting methods, researchers applied SVM, and MLP to classify the negative emotions. At the end of the study, researchers observed that the best accuracy was obtained with MLP. In the study of [29], the authors proposed a method for feature extraction by using DTCWT (double tree complex wavelet transform). In the study, 16 subjects and visual stimuli were used. In the classification part, the SVM algorithm was applied, and six different emotions were classified. The F3, FP2, C5, C6, P3, and P4 channels were selected in the analysis and normalization of the data. Finally, they got an accuracy of 90.61%.

In line with the studies examined, it has been seen that artificial intelligence-based approaches are effective in the analysis of emotion. Due to these achievements, deep learning, which is one of the artificial intelligence methods, was used in this study and the classification of emotions was carried out.

## 3. Material and Methods

### 3.1. GAMEEMO Dataset

In this study, the GAMEEMO dataset is employed [19]. The data set contains EEG signals of 28 subjects. The length of the EEG signals in this data set is 38,252. The age range of the subjects is between 20 and 27 years old. This dataset includes computer games-based EEG signals. They are collected from 28 different subjects with wearable and portable EEG device called 14 channel Emotiv Epoc+. Subjects played 4 emotionally different computer games (boring, calm, horror, and funny) for 5 min, providing a total of 20 min worth of EEG data available for each subject. Unlike other data sets in the literature, computer games were used as stimuli in this data set. In addition, the data were obtained with a portable EEG device. The device contains 14 EEG channels including AF3, AF4, F3, F4, F7, F8, FC5, FC6, O1, O2, P7, P8, T7, and T8. Since the data were obtained with a portable EEG device, connections with the subjects were made via Wi-Fi. While generating the dataset, both

discrete emotion model and dimensional emotion model were taken into consideration. In the study, a total of 4 games were played to capture emotions and these games were labeled as G1, G2, G3, and G4. In the discrete emotion model structure, G1 and G3 express negative emotions, while G2 and G4 express positive emotions. For the arousal-valence dimensional model, G1 refers to LANV, G2 refers to LAPV, G3 refers to HANV and G4 refers to HAPV. Table 1 explains the details of GAMEEMO data set. The sampling rate of the signals is 128 Hz. The dataset contains both raw EEG data and pre-processed EEG data. More detailed and technical information about the data set can be found in the study [19].

**Table 1.** The stimuli that used in the GAMEEMO data set.

| Game Symbol | Stimuli Type | Discrete Model | Dimensional Model |
| :---: | :---: | :---: | :---: |
| G1 | Boring | Negative emotion | LANV zone |
| G2 | Calm | Positive emotion | LAPV zone |
| G3 | Horror | Negative emotion | HANV zone |
| G4 | Funny | Positive emotion | HAPV zone |

In this study, pre-processed EEG data were used. The data acquisition stage is as follows:

- In the first stage, four different computer games were played by the subjects. The subjects played each game for 5 min. In total, EEG data of 20 min was obtained for each subject.
- The subjects filled out the SAM (Self-Assessment Manikin) form after playing each game. The main purpose of filling out this form is to label each game played.

### 3.2. Feature Extraction Methods

EMD and VMD were used for feature extraction in the study. While 9 IMFs were obtained with EMD, a total of 5 IMFs were obtained with VMD. The main reason for collecting 9 IMFs with EMD and 5 IMFs with VMD is that the EEG signals are converted into this number of IMFs at most. Due to the properties of the EEG signals, a maximum of 9 IMFs were obtained by EMD. This situation is calculated as 5 with VMD. Afterward, statistical features were obtained from these 14 IMFs collected. These obtained statistical features are maximum value, minimum value, and average value.

#### 3.2.1. Empirical Mode Decomposition

EMD was first introduced at the end of the 1990s. This method is based on analyzing the original signal into a subset of signals repeatedly and in different patterns depending on separate spectral bands [30,31]. This algorithm is used a lot, although it does not contain an accurate mathematical equation, and it faces difficulties in separated similar frequencies. As this algorithm repeatedly detects the minimum and maximum limits in the signal, it also removes the average of the envelopes as a low pass center line [31]. The power of this algorithm has decreased since it has become highly dependent on methods of extreme point finding, interpolation of extreme points into carrier envelopes, results of neuroscience experiments, electro diagrams, and sea-surface height readings in analyzing the signal [30]. Despite this mathematical deficiency, this algorithm had a great impact on signal analysis and has been applied a lot in biomedical signal processing. Studies, such as signal analysis in sound engineering [32], climate analysis [33], and epilepsy detection [34], can be given as examples. Due to its success in biomedical fields, the EMD method was used in this study to decomposes a signal *x(t)* into a series of IMFs, as in Equation (1) [35].

$$x(t) = \sum_{j=0}^{i} I_{mode}(j) + I_{res} \tag{1}$$

where $I_{mode}(j)$ represents IMF, $I_{res}$ specifies a low-order polynomial component. After applying this equation, a limited number of intrinsic mode functions will be produced

nearly perpendicular to each other. Figure 3 shows the IMFs of the EEG signal used in this study, obtained by EMD.

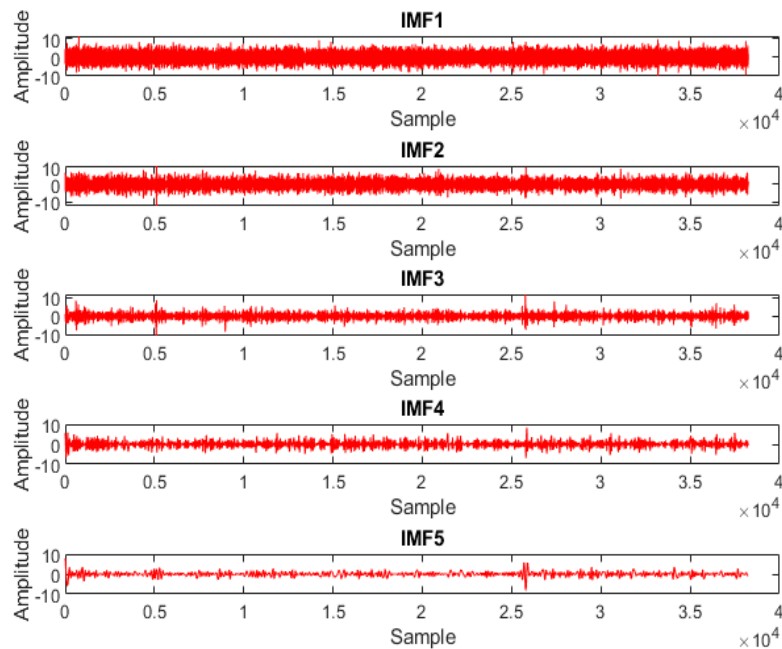

**Figure 3.** First five IMFs using EMD of the F4 EEG channel belonging to the subject number 10 (according to the G1 label).

3.2.2. Variation Mode Decomposition

VMD performs time-frequency decomposition like EMD, by decomposes the signal into $k$ discrete number of sub-signals [36]. It has been developed to overcome the limitations arising in EMD [37]. Since VMD does not use a recursive approach, it obtains IMFs with a concurrent approach. Furthermore, it is an adaptive approach, and it converts the signal into a certain number of IMFs with the relevant center frequencies. IMFs are the AM-FM (amplitude modulated frequency modulated) signals, and they can be calculated using the Equation (2) [37].

$$uk(t) = A_k(t)\cos(_k(t)) \tag{2}$$

where $_k(t)$ represents the non-decreasing function and the derivative of the $_k(t)$ (instantaneous frequency) and should be more than or equal zero. $A_k(t)$ represents the amplitude and it is more than or equal zero as well. Finally, $uk(t)$ specifies the sub-signals. IMF also allows non-discontinuous and discontinuous signals like the sawtooth. After calculating the mode, the bandwidth of the mode is determined based on Carson's rule, as in Equation (3) [37].

$$BW_{FM} = 2(\Delta f + f_{FM}) \tag{3}$$

where $BW$ refers to the practical bandwidth, $\Delta f$ represents the maximum deviation, and $f_{FM}$ shows the carrier frequencies. In addition, Equation (4) is used to calculate the total practical IMF bandwidth, and Equation (5) is used to calculate the VMD method.

$$BW_{AM-FM} = 2(\Delta f + f_{FM} + f_{AM})) \tag{4}$$

$$\min_{u_k,w_k}\left\{\sum_k \|\vartheta_t\left[\left(\left(\delta(t) + \frac{j}{\pi t}\right) * u_k(t)\right)e^{-jw_k t}\right]\|_2^2, \sum_k u_k = f(t)\right\} \tag{5}$$

where $f(t)$ represents the signal, $u_k$ specifies the signals' mode. $w_k$ shows the central frequency, $\delta$ represents the Dirac distribution depicting the noise tolerance. Furthermore, $\vartheta_t$ denotes the time derivative, $(t)$ specifies the time script. Finally, $(k)$ shows the number of

IMFs. Compared to EMD, the VMD method was found to be more effective against noisy signals [38]. When the literature is reviewed, it is seen that VMD is used effectively in the biomedical field at least as much as EMD. Emotion recognition [22], EMG-based signal studies [30], and diagnosis of lung disease [31] can be given as examples. The effective use of the VMD method in the biomedical field has enabled us to use this method in this study. Figure 4 shows the IMFs of the EEG signal used in this study, obtained by VMD.

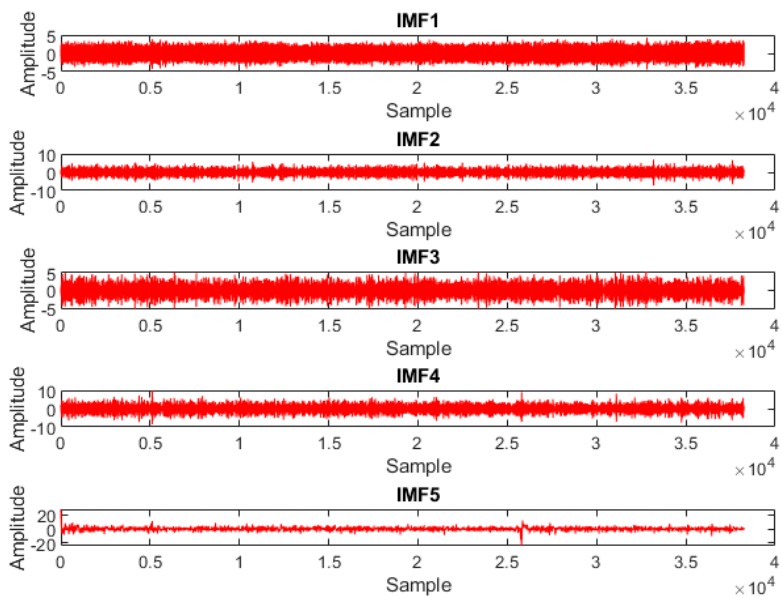

**Figure 4.** First five IMFs using VMD of the F4 EEG channel belonging to the subject number 10 (according to the G1 label).

Statistical features were extracted from a total of 14 IMFs obtained in EMD and VMD. These statistical features are the maximum value, the minimum value, and the average value. We believe that the features obtained by using statistical methods give acceptable results and are still used in the signal processing field. There are several types of statistical methods, such as min, max, average, median, standard deviation, etc. We selected three of them (maximum, minimum, and average). The maximum and minimum values show the start and endpoints of the identified EEG signals, with the maximum and minimum value showing the maximum amplitude and minimum amplitude values of the samples of the EEG data. These values are required to compare the highest and lowest peak signals in different mental states. In the region selected for sample data, the peak-to-peak (P-P) shows the difference between the value of the maximum amplitude and the minimum amplitude. Average computes the average amplitude value of the EEG data samples collected between endpoints in the selected region. Equation (6) is used to extract the mean value of the EEG signal [39]:

$$Avg = \frac{1}{n_e - n_s} \sum_{i=n_s}^{n_e - n_s} X_{iEEG} \tag{6}$$

where $n_s$ refers to the starting point of the signal, $n_e$ refers to the endpoint of the signal. Besides, $n_e - n_s$ represents to the total number of samples, and variable $i$ refers to the values on the horizontal axis.

### 3.3. Classifier

In this study the BiLSTM network was presented as a deep learning network for detecting emotions. In this network the output *(t)* is based on the previous segment *(t − 1)* and next segment *(t + 1)*. The general concept of a proposed multilayer BiLSTM network is shown in Figure 5 [40]. This network can be effectively used for processing time series

data, and it is faster and more accurate than traditional LSTM, RNN (recurrent neural networks) and GRU (gated recurrent units) [41]. Furthermore, a BiLSTM layer learns bidirectional long-term dependencies between time steps of time series or sequence data. These dependencies can be useful when the network is about to learn from the complete time series at each time step [42,43]. In addition to these, traditional LSTM and RNN only use information from the past since the only inputs it has to consider are from the past. Yet, BiLSTM runs inputs in two ways, including from past to future and from future to past. In other words, the flow of information is in two directions. While this ensures that information flows in both directions in BiLSTM architectures, it causes it to flow in one direction in traditional LSTM architectures. The LSTM and RNN architectures work backwards and use only information from the future. This causes the loss of information and causes the method to be more ineffective compared to the BiLSTM architecture [44].

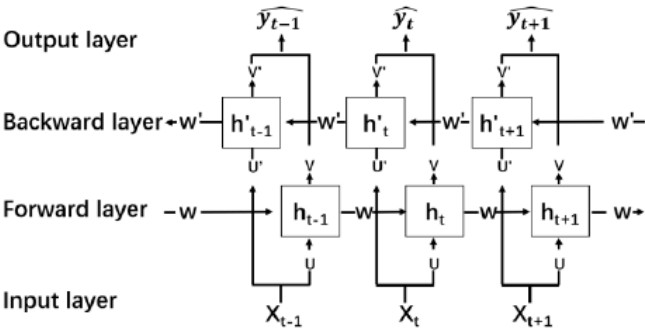

**Figure 5.** The structure of BiLSTM network.

*3.4. K-Fold Validation*

Cross validation is an extensive method of using all available data as examples for training and testing, to prevent the overfitting, where all available data are divided into identical small groups according to parameter *k*. Figure 6 explains the method of data splitting in the k-fold technique. For example, if the value chosen for the parameter *k* is 10, then the data will be divided into 10 equal groups, the first one for testing and the rest for training. Then, the second for testing and the rest for training, and so on, until we get to the tenth group [45].

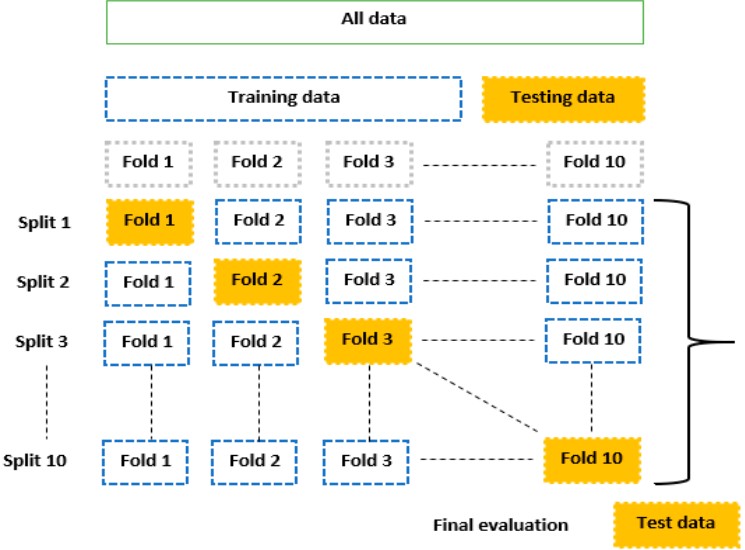

**Figure 6.** The k-fold technique that used in the study.

## 4. Application Results and Discussion

In this study, the data belonging to the GAMEEMO dataset were used and emotion analysis was performed by classifying both discrete emotion model, and dimensional emotion model. Three machine learning and one deep learning algorithm were used for the classification process. The kNN algorithm is an algorithm that works based on supervised learning. It is simple and easy to implement. It is generally used effectively in classification and regression problems [46]. The algorithm is basically based on similarity between neighbors. It is a method based on transferring data that are close to each other to the same class or cluster. Each new sample that comes to the data space is compared with the data in the dataset and placed in the class or cluster it belongs to according to similarity. kNN used in various fields such as data mining, pattern recognition and security. One of the biggest advantages of the kNN algorithm is that it gives successful results on data containing noise. Furthermore, it has been observed that as the number of data increases, the algorithm gives successful results [47]. Among the disadvantages, it can be shown that the success of the classification varies depending on the selected distance calculation method and the *k* value [47]. As the distance between data is calculated one by one, there may be a decrease in the performance of the classification. SVM is a machine learning algorithm based on statistical learning theory and developed with the technique of supervised learning. It was first introduced in 1995 and aimed to solve classification problems with pattern recognition [48]. In the early days, it was used only to distinguish binary-class linear data, but nowadays it can be used to distinguish both non-linear and multi-class data. The basis of the classification process with SVM lies in the separation of data belonging to two different classes with a specified boundary, namely the hyperplane. When the data are not linearly separated, nonlinear SVMs are used. Various kernels are used for this, such as radial base, polynomial, and sigmoid. SVM is effective in situations where the distinction between data is clear [49]. Similarly, it is effective in higher dimensional spaces. On the other hand, if the dataset is large, it is not an appropriate classification algorithm [49]. RF is a classification algorithm formed by combining multiple decision trees. Each individual tree in the random forest spits out a class prediction and the class with the most votes become the model's prediction. The RF algorithm is mentioned as one of the ensembles learning methods. Today, it is used effectively in many areas and produces considerable results as a classification performance. One of the biggest advantages of RF is that it runs effectively on large datasets [50]. In addition, it can predict which variable is more important in a classification process. It produces successful results in predicting missing data. Furthermore, as with every classifier, it also has disadvantages. One of its biggest limitations is that the algorithm works slowly when the number of trees is high, and it is ineffective in real-time prediction. Another cutoff is that the training time is long. This situation is caused by the large number of trees in the algorithm. In addition to machine learning algorithms, one deep learning has been designed and used in this study. The designed deep learning method is based on bidirectional long-short term memory architecture. BiLSTMs can be expressed as an extension of traditional LSTM architectures. The main purpose of developing this architecture is to improve the performance in sequence classification [51,52]. BiLSTMs train two LSTMs instead of one in the input sequence. The first on the input sequence as-is and the second on a reversed copy of the input sequence. This can provide additional context to the network and result in faster and even fuller learning on the problem. The fact that learning could be easier and faster led us to focus on BiLSTM in this study. In addition, keeping and using past information is one of the biggest advantages of BiLSTM. Due to these advantages, BiLSTM was used in this study. If a classification model typically contains at most one or two layers of non-linear feature transformations, it is considered as shallow architecture [53–55]. We used a three-layer BiLSTM unit so that the model we designed was not shallow. In this study, a deep BiLSTM (DeepBiLSTM) architecture is designed using the BiLSTM architecture. The developed DeepBiLSTM architecture was created by combining three consecutive BiLSTM architectures and the parameters of the developed DeepBiLSTM model are given below:

- A total of 1568 (14 × 28 × 4) pieces of 4704 (14 × 3 × 28 × 4) sample length EEG signals were used in the input layer. Above, 14 refers to the number of EEG channels, 28 refers to the number of subjects, 3 indicates the collected statistical features, and 4 represents the zones of the arousal-valence plane.
- 256-unit BiLSTM was used in the second layer. ReLU (rectified linear unit) was preferred as the activation function.
- BiLSTM with 128 units was preferred for the third layer. As in the previous layer, ReLU is employed as the activation function in this layer.
- A total of 64 BiLSTM units were used in the fourth layer. As with the previous two layers, ReLU was used as the activation function.
- Then, flattening was carried out so that all data were one-dimensional.
- After flattening, batch normalization has been performed so that all data are in the same range (between 0 and 1).
- Then, dropout was conducted to prevent excessive learning and overfitting and 15% of the neurons were removed from the architecture.
- Two fully connected layers were considered. While 512 neurons were used in the first fully connected layer, the number of neurons was reduced to 256 in the second fully connected layer.
- In the last layer, a classification process was made. Two neurons were used in the last layer for binary-class classification. For the multi-class classification process, in the last layer, 4 neurons were used as the number of classes is 4.
- For the binary-class classification process, Sigmoid is used as the activation function in the classification layer and Softmax is used as the activation function in multi-class classification.
- The error of the model is determined by the binary cross-entropy function for binary class classification. In the multi-class classification process, the error of the model is determined by categorical cross-entropy.
- The Adam function was used for optimization in both classification processes. The learning rate of Adam function was defined as 0.001 and the decay value was 0.00001.
- The number of epochs of the model was determined as 250.
- Validation of the model was carried out with 10-fold cross-validation.
- All these parameters were determined by a trial-and-error approach.

In addition to these, the parameters of SVM, RF and KNN algorithms were determined as follows. As in the DeepBiLSTM, the parameters in these classifiers were determined by trial-and-error approach. The parameters of the SVM classifier are as follows:

- Linear kernel was used as the kernel of the model for binary-classification. For multi-class classification purpose, GRB (Gaussian Radial Basis) kernel was applied.
- Size of the margin value (C) is defined as 1.5.

The parameters of the RF classifier are as follows:

- The maximum depth of the model was determined as 25.
- A total of 500 estimators were used.

The parameters of the KNN classifier are defined as:

- In total, three neighbors were used and the distance between neighbors was calculated with Euclid.
- The weight of the model was determined by the 'distance' parameter.
- Brute was used for the algorithm of the model.
- The value of 20 was taken into account as the leaf size.

The performance of the classifiers is determined by accuracy and ROC. The ROC provides an illustration of the ratio between sensitivity (true positives) and specificity (true negatives). The ROC analysis provides important information about the performance of the diagnostic test as if the vertex of the curve is close to the upper left corner, the sensitivity rate (true positives) is high and the specificity (true negatives) is low [56,57]. AUC (area under curve) is also a measure of the discriminative strength of the test. If the AUC value

is 0.5, it means that the sensitivity is 50% and the specificity is 50%, meaning there is no differential value. Studies indicate that an AUC of less than 0.75 is not clinically useful and an AUC value of more than 0.90 has a very high clinical value [56]. Figure 7 shows the flow chart of the study. As can be seen from Figure 7, four classifiers were used, and the performance of these classifiers was measured by the accuracy score. The classification results obtained using the proposed method and various machine learning algorithms are given in tables.

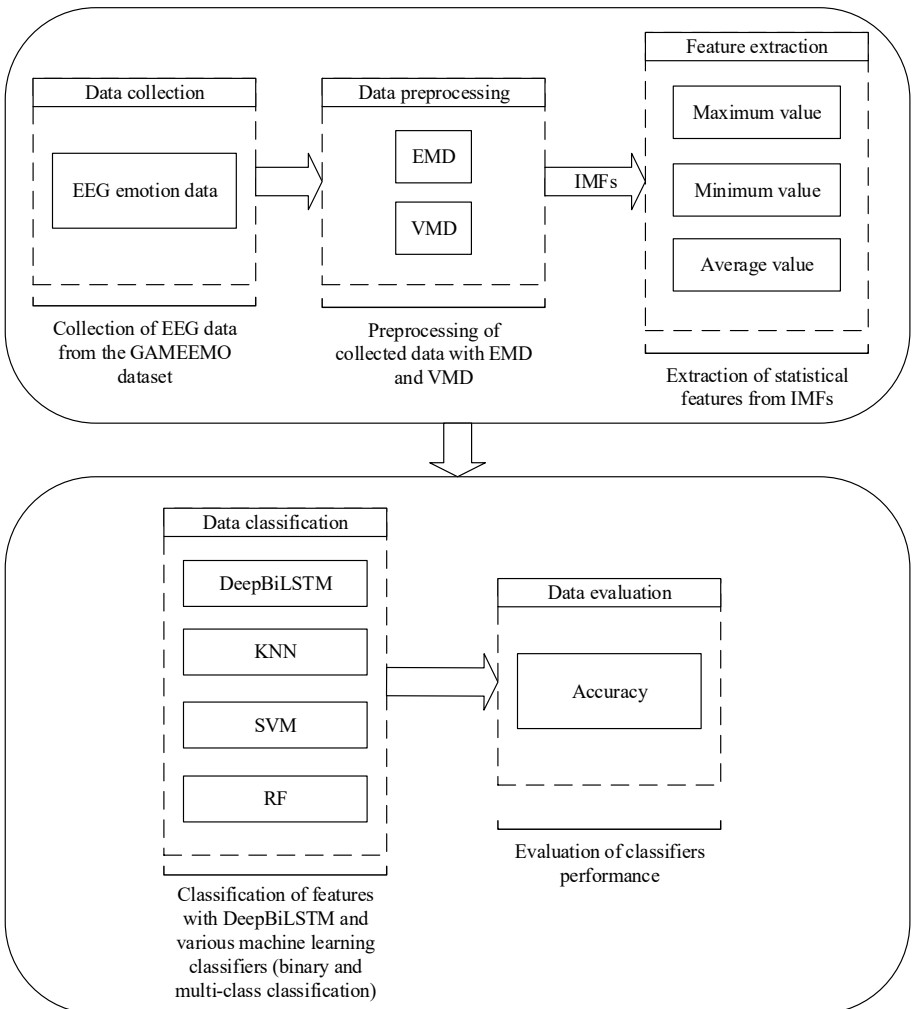

**Figure 7.** Flow chart of the study.

In the first step (data collection), EEG signals of the GAMEEMO data set were obtained. There are 14 channels in total in the data set and each channel contains 38,252 time series. After the data were obtained, EMD and VMD methods were applied to the EEG signals in the second step (data preprocessing). In this way, the size of the data has been reduced and made suitable for classification. A total of 14 IMFs were obtained, nine IMF for EMD, and five IMF for VMD. In the third step (feature extraction) after the preprocessing, feature extraction was done, and only statistical features were used for this. In the feature extraction process, the maximum, minimum and average values were calculated from the obtained 14 IMFs and the feature vector was created. In the fourth step (data classification), features were classified and SVM, KNN, RF, and BiLSTM classification algorithms were used accordingly. The classification process was carried out for both binary-class and multi-class. While only positive-negative emotions were used in the binary classification, the dimensional model was used in the multi-class classification. HAPV, LAPV, HANV,

and HAPV zones were considered in the multi-class classification process. In the last step (data evaluation), the performance of the classifiers was evaluated and only the accuracy score was used.

The accuracy and some statistical results of the binary-class classification process are shown in Table 2. In Table 3, the accuracy values and some statistical information of the multi-class classification process are given.

**Table 2.** Accuracy, mean and standard deviation results of positive-negative emotions.

| Fold | kNN | SVM | RF | DeepBiLSTM |
|------|------|------|------|------|
| F1 | 64.58% | 51.89% | 54.23% | 69.14% |
| F2 | 63.25% | 55.78% | 51.27% | 71.42% |
| F3 | 60.24% | 56.87% | 56.41% | 70.83% |
| F4 | 64.58% | 51.23% | 53.62% | 69.50% |
| F5 | 66.47% | 56.41% | 51.24% | 72.48% |
| F6 | 62.96% | s53.87% | 55.68% | 70.20% |
| F7 | 64.12% | 58.97% | 58.00% | 70.00% |
| F8 | 62.24% | 55.87% | 55.29% | 69.60% |
| F9 | 66.39% | 55.26% | 58.74% | 71.22% |
| F10 | 69.87% | 58.79% | 64.26% | 74.52% |
| Mean | 64.47% | 55.49% | 58.74% | 70.89% |
| Standard Deviation | 2.65% | 2.58% | 3.86% | 1.63% |

**Table 3.** Accuracy, mean and standard deviation results of arousal-valence emotions.

| Fold | kNN | SVM | RF | DeepBiLSTM |
|------|------|------|------|------|
| F1 | 66.32% | 62.35% | 65.85% | 89.40% |
| F2 | 61.47% | 60.47% | 72.45% | 87.14% |
| F3 | 63.63% | 62.84% | 71.48% | 81.48% |
| F4 | 68.97% | 69.87% | 70.32% | 87.44% |
| F5 | 67.89% | 63.12% | 74.87% | 95.65% |
| F6 | 66.21% | 61.28% | 68.95% | 88.51% |
| F7 | 65.87% | 67.74% | 69.41% | 95.95% |
| F8 | 67.84% | 67.45% | 70.32% | 86.42% |
| F9 | 67.41% | 64.05% | 73.82% | 93.86% |
| F10 | 69.44% | 68.44% | 67.44% | 97.44% |
| Mean | 66.40% | 64.76% | 70.49% | 90.33% |
| Standart Deviation | 2.43% | 3.32% | 2.78% | 5.16% |

When the results in Table 2 are examined, generally, it can be seen that the classifiers have not performed very well. The best performance was obtained with the proposed deep learning, and the classification between positive and negative emotions was made with an accuracy rate of 70.89%. The closest performance to the proposed deep learning model was achieved with the kNN algorithm. With SVM and RF, the accuracy values remained under 60% for each fold. The reason why the results were not successful is due to the distribution of the data.

In Table 3, the accuracy results of four zones belonging to the arousal-valence plane are given. According to the results in Table 3, it can be observed that the multi-class classification process is more successful than the binary-class classification. The accuracy value of SVM and RF algorithms increased in multi-class classification. The performance of the SVM algorithm in multi-class classification has increased to 64.76%. Similarly, the performance of the RF algorithm has increased noticeably, and the average accuracy has increased up to 70.49%. The accuracy performance of the kNN algorithm has also increased, but no significant increase has been observed as in SVM and RF. Furthermore, looking at the accuracy result of the proposed deep learning model, it has been observed that the proposed method produces a much better result compared to other methods. The proposed method was more effective than other methods in both binary-class classification

and multi-class classification. One of the biggest reasons for this situation is that the data distribution in multi-class classification is more distinctive than binary-class classification. In Figure 8, the data distribution of positive–negative emotions are given. As can be understood from Figure 8, positive and negative emotions show intensity in a certain area. The close proximity of the data can make the classification process difficult. Looking at the distribution of data, it is not surprising that the results are not good enough. The fact that the values of positive and negative emotions were close to each other caused the classification process to be not reliable enough. We have previously stated that the SVM algorithm is effective when data distribution is clean. This performance of SVM confirms this conclusion. kNN algorithms are more effective in data containing more noise. The ineffectiveness of the algorithm may be since these data do not contain noise. We have stated that the RF algorithm is also effective in large datasets. Our insufficient number of data may have caused this classifier to be ineffective. However, even in this data distribution, the proposed deep learning method has made a remarkably successful classification. This confirms the success of BiLSTM in sequence classification. The main purpose of distinguishing positive and negative emotions is to determine the performance of the proposed method in binary classification. In addition, the number of data is increased by performing binary classification. In this way, the training of the proposed method is made more robust.

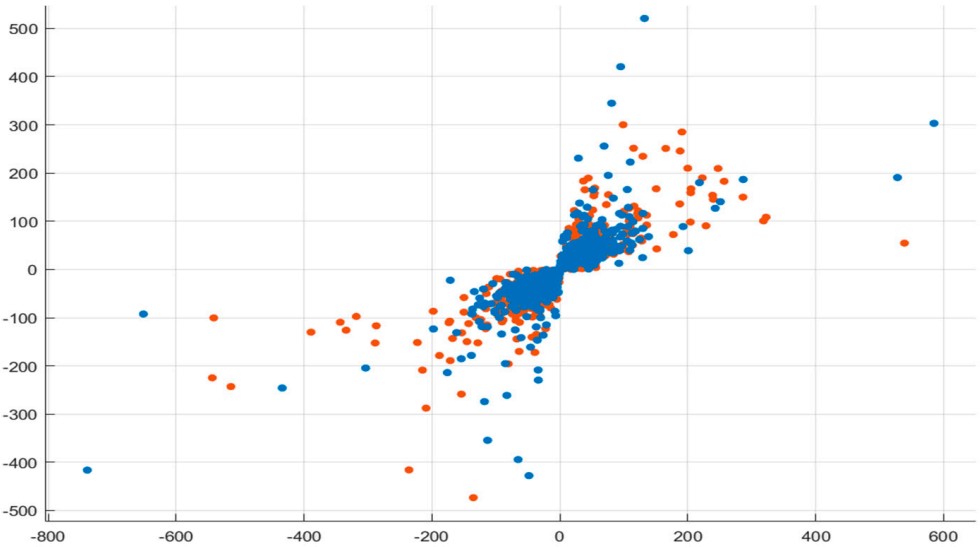

**Figure 8.** Data distribution of positive and negative emotions. (Blue dots signify negative emotions; red dots signify positive emotions).

The data distribution of the multi-class is given in Figure 9. Looking at the data distribution in Figure 9, it is seen that the distribution between points is more uniform than the binary-class classification. Differences in emotions are less than binary-class classification. The classification results also confirm this situation. The more decisive the distinction between the points, the more successful the classifications. One of the biggest reasons for the improved performance of the SVM classifier is that the distinction between these points is clear. kNN has not improved its performance much compared to binary-class classification. One reason for this may be that the data are not suitable for kNN (they do not contain noise), as in binary-class classification. One of the biggest reasons for the increase in the performance of the RF algorithm is the increase in the number of labels. In this way, the number of classes has increased, which significantly affects the performance of the algorithm.

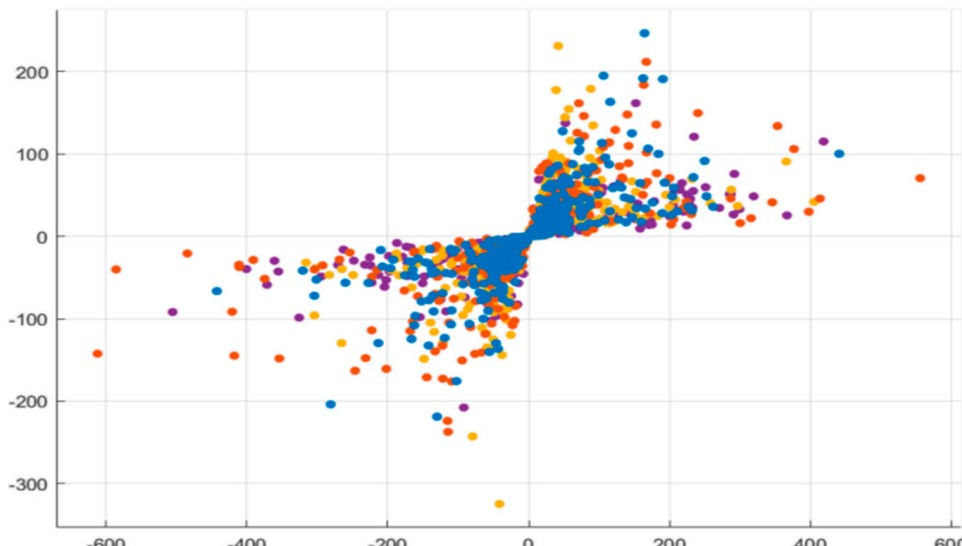

**Figure 9.** Data distribution of emotions in the arousal-valence plane. (Blue dots indicate LANV; red dots refer LAPV; yellow dots indicate HANV; and purple dots show HAPV emotions).

As with every study, this study as certain advantages and disadvantages. The advantages can be listed as follows:

- It has been observed that the deep learning algorithm is better than certain machine learning algorithms, despite the small number of data. Generally, the number of data is expected to be high in studies conducted with deep learning.
- Although the data distribution is not very clear in binary classification, the proposed deep learning model has been observed to be acceptably successful.
- When classifying with deep learning, manual feature extraction is generally not performed. In this study, we performed deep learning after feature extraction. By extracting features from signals, we may have caused certain information to be lost. However, despite this, a successful classification was achieved by the proposed deep learning model.
- A lot of information cannot be obtained from statistical features. Generally, effective features in signal processing studies are obtained from time-frequency features [54,55]. Despite this, the deep learning model has produced successful results.

The disadvantages of the study can be listed as follows:

- Due to the large sample length of the signals in the data set, we could not process the raw data. We had to perform feature extraction. This may have caused some information to be lost. Processing the raw data can increase the performance of the deep learning algorithm.
- The performances of the deep learning model and other machine learning models vary according to the selected feature extraction methods. More information can be obtained by using other feature extraction methods. This can positively affect the performance of the methods.
- Although the number of data is small, the proposed method has been successful. However, this does not mean that the model is reliable. To determine the reliability of the model, different data sets should be used, or the information obtained from the data set used should be reproduced.

Figure 10 shows the ROC curve for binary classification, with both classes using Deep-BiLSTM. According to Figure 10, the accuracy jumped from 70.89% into 78%. Figure 11 shows the ROC curve for multi-classification, with all classes using DeepBiLSTM. According to Figure 11, the accuracy decreased from 90.33% to 89%.

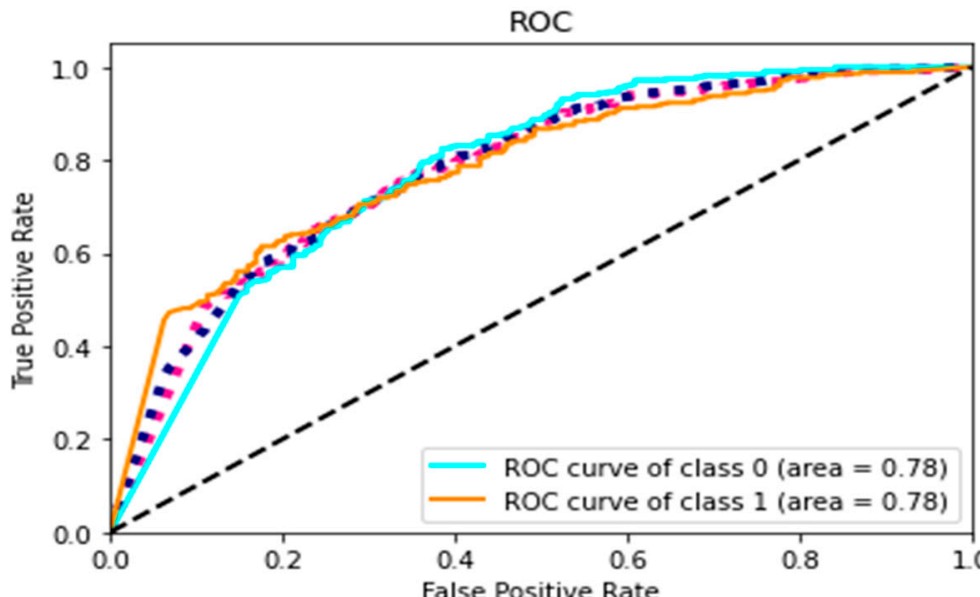

**Figure 10.** The ROC curve for binary classification using DeepBiLSTM. (Pink squares represent micro-average ROC curve, while purple squares represent the macro-average ROC curve).

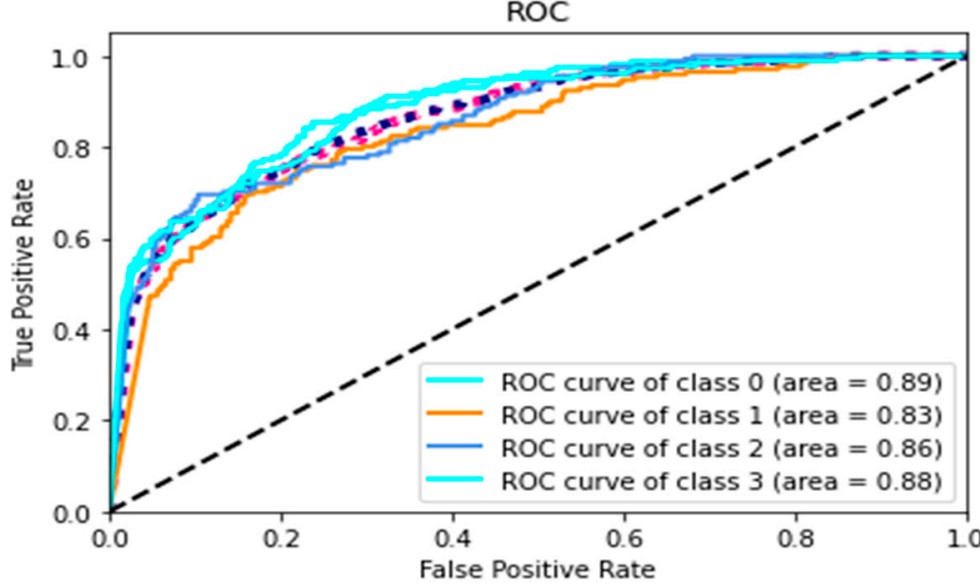

**Figure 11.** The ROC curve for multi-class classification using DeepBiLSTM. (Pink squares represent micro-average ROC curve, while purple squares represent the macro-average ROC curve).

In addition, the obtained results were compared with various studies in the literature. The results of this comparison are given in Table 4.

Looking at the results in Table 4, it was observed that the results obtained in this study were at least as effective as the aforementioned methods. In addition to these, the results of other studies using the GAMEEMO dataset and deep learning algorithms in the literature are given in Table 5. In this way, the proposed method has been compared with other deep learning algorithms.

**Table 4.** A comparison table with previous studies.

| Ref. | AI Methods | Emotion Model | Feature Extraction Methods | Performance Criteria | Results (Avg.) |
|---|---|---|---|---|---|
| [19] | MLPNN, SVM, kNN | Discrete and dimensional | Statistical features, DWT, Hjorth features, Shannon entropy, logarithmic energy entropy, sample entropy, multi-scale entropy | Accuracy | 80% |
| [20] | BiLSTM, kNN, SVM, ANN | Discrete model | Spectral entropy | Accuracy | 76.91% |
| [21] | kNN, SVM, ANN | Discrete model | DWT, EMD | Accuracy | 94.3% |
| [22] | DNN | Dimensional model | VMD | Accuracy | 61.88% |
| [57] | kNN, NB, DT, CNN | Dimensional model | EMD, VMD, entropy, HFD (Higuchi's Fractal Dimension) | Accuracy | 95.20% |
| [58] | SVM | Discrete model | EMD, VMD | Accuracy | 90.63% |
| [23] | kNN, LR, NB, SVM | Discrete model | ANOVA | Accuracy | 95.45% |
| [24] | SVM, kNN | Discrete model | PSD (Power Spectral Density) | Accuracy | 99.42% |
| [26] | SVM | Discrete model | HOG, GC, TE | Accuracy | 95.21% |
| **This study** | **SVM, kNN, RF, DeepBiLSTM** | **Discrete and dimensional model** | **EMD, VMD, statistical features** | **Accuracy** | **80.81%** |

**Table 5.** Comparison of the designed deep learning model with other deep learning models.

| Ref. | AI Methods | Results (Avg.) |
|---|---|---|
| [59] | BiLSTM | 76.93% |
| [60] | GoogleLeNet | 93.31% |
| [61] | DEEPHER | 99.99% |
| [62] | LSTM | 96.00% |
| [63] | CapsNet | 98.89% |
| [64] | RNN + LSTM | 98.44% |
| [65] | LEDPatNet19 | 99.29% |
| **This study** | **DeepBiLSTM** | **90.33%** |

When the results in Table 5 are examined, it is seen that the designed BiLSTM method is at least as effective as the methods in the literature. However, the proposed method was weaker compared to CNN-based approaches. Images are used in the CNN-based classification process and features are obtained from the images. It is not surprising that the results are more effective in CNN-based approaches. One of the most important reasons for this is the low loss of information while processing the image. Furthermore, its built-in convolutional layer reduces the high dimensionality of images without losing its information. However, based on the results of the classification process made according to the RNN structure, it has been seen that the proposed method is the second most

effective method. While 96% accuracy was obtained in the classification process with LSTM, 76.93% accuracy was obtained in the classification process with BiLSTM. With the proposed method, 90.33% accuracy was calculated. Using different feature extraction methods and increasing the data size can increase the result of this classification. However, since one of the aims of the study was to make an effective classification with a small number of features, the classification result obtained was at least as effective as LSTM. As in this study, BiLSTM architecture was used in study [20] and accuracy of 76.93% was obtained in that study. In that study, features were not extracted from the EEG signals. Raw EEG signals are given directly to the BiLSTM architecture, and the data are classified. In addition to these, single-layer BiLSTM architecture was used in that study. In this study, unlike the study [20], the raw signals were first converted into sub-signals with VMD and EMD, and a total of 14 IMFs were obtained. Then, statistical features were obtained from these 14 IMFs and classification was made using these features. Finally, in this study, a three-layer BiLSTM architecture was used, and the network was provided with a deep structure. As a result, the depth structure of the deep learning model used and the process of obtaining the features are different from the study [20]. These improvements have increased the classification result from 76.93% to 90.33%.

The limitations of existing studies can be expressed as follows:

- The concept of emotion is an abstract structure. It varies from person to person. Therefore, it is not possible to reach a definite conclusion no matter what study is done.
- In current studies, visual or aural stimuli are generally used separately. Using them separately does not have a sufficient stimulating effect.
- In some other studies in the literature, visual and aural stimuli are used at the same time, and videos or music clips are usually shown for this purpose. However, these stimuli are not as effective as computer games.
- Traditional EEG devices are generally used in current studies. This device is both difficult and costly to use.

The novelties of the study can be expressed as follows:

- In this study, it was observed that emotion analysis based on EEG signals was more effective. Because EEG signals cannot be manipulated by the subjects and real emotions cannot be hidden.
- In this study, it was observed that the classification made according to the emotion model (arousal-valence) was more effective than the discrete emotion model (positive-negative). This showed that as the number of classes increases, the classification process is healthier, and the deep learning model learns better.
- Raw EEG signals were used in the study and no specific sampling was performed. In this way, the process load was less, and time was saved.
- With this study, it has been observed that portable EEG devices are at least as effective as conventional devices. We think that the results of this study will encourage researchers to use portable devices.
- In the study, only statistical methods were used for feature extraction and the data size was not very complex. Only maximum, minimum, and average values were used for feature extraction and the size of the data was reduced. This situation did not adversely affect the classifier and an effective classification was performed. In this way, it has been observed that few and concise features are effective in classification rather than obtaining a large number of features.
- It has been observed that the deep learning algorithm is more effective than some of the machine learning algorithms.

## 5. Conclusions

In this study, emotion analysis was performed using the EEG data of the GAMEEMO dataset. In the study, both the discrete emotion model and the dimensional emotion model were classified. The study consisted of four stages: In the first stage, EEG data were

obtained from the dataset. Then, EEG signals were transformed into sub-signals with EMD and VMD methods and a total of 14 IMFs were obtained, including nine from EMDs and five from VMDs. Later, feature extraction process was performed, and statistical features were obtained from these 14 IMFs. Statistical features obtained in the study are maximum value, minimum value and average value. After the feature extraction process, the features were classified with both various machine learning algorithms and the designed deep learning algorithm. The performances of the classifiers were measured with the accuracy score and compared. It was observed that the proposed method performed best in both binary-class classification and multi-class classification. SVM and RF yielded the worst accuracy results in binary-class classification. In contrast, kNN was more effective than these two methods and reached the closest accuracy to the proposed deep learning model. The best performance was achieved with the proposed DeepBiLSTM deep learning model and an accuracy of 70.89% was achieved. In the multi-class classification process, there were noticeable increases in the accuracy performance of SVM, RF and recommended DeepBiLSTM models. In the binary-class emotion analysis process, SVM and RF algorithms did not exceed 60% accuracy, yet they achieved 64.76% and 70.49% accuracy, respectively, in the multi-class classification process. While there was no considerable increase in the kNN algorithm, an effective classification process was performed with the proposed DeepBiLSTM model. In the multi-class analysis process, 90.33% accuracy was achieved with the proposed DeepBiLSTM model. In this way, the effect of the proposed deep learning model was demonstrated. The success of this method in this area will be observed in other datasets and the contribution of the model to the literature will be provided in future studies. In future studies, using different types of portable EEG devices, the performance of such devices in this field will be improved. In addition to these, the number of data will be increased by playing different types of computer games and the effect of this type of stimulus will be determined more clearly. Finally, in future studies, an application based on human–computer interaction that can determine emotion analysis in real-time will be implemented.

**Author Contributions:** Conceptualization: M.B.; methodology: T.B.A., A.A. and M.B.; software, T.B.A. and M.B.; formal analysis: A.A. and M.B.; writing-original draft preparation: A.A. and M.B.; writing-review and editing: T.B.A.; supervision, M.B.; funding acquisition, M.B. All authors have read and agreed to the published version of the manuscript.

**Funding:** This research has received no external funding.

**Institutional Review Board Statement:** Not applicable.

**Informed Consent Statement:** Not applicable.

**Data Availability Statement:** All information about the data used in this study are explained in the third section.

**Conflicts of Interest:** The authors declare no conflict of interest.

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
