# Peer review of "A Novel Approach for Emotion Recognition Based on EEG Signal Using Deep Learning"

_applsci, doi:10.3390/app121910028_

Round 1

Reviewer 1 Report

In this paper, there is no novelty though the presentation of the paper is good.

Mostly the authors have used BiLSTM to conduct the experiment and got some results.

there is no such proper evidence behind doing so, as they didnot realize about the complexity of the model.

In table 2, comparision is not made among the similar level of classifiers rather weak classifiers with the proposed method.

In fig 8, it is clearly visible on the splitting of points which demonstrates about the variations on the data points.

I didnot find the sampling process of data and as a large amount of processed data, it is quite ovious to have oversampling.

Author Response

Thank you for your valuable comments.We have sent the replies to your comments as attachments. Thank you for evaluating our article. Please see the attachment. 

Reviewer 2 Report

Well written and structured research. The authors conduct a quantitative study supported by experimental data with modern Approach for Emotion Recognition Based on EEG Signal. They use modern equipment to conduct the research - EEG listeners. It was made among a young audience between 20-27 years old. Artificial intelligence and machine learning methods were used to analyze the results. The quantity and quality of the literature used are sufficient.

I only recommend checking compliance with the publication template when finalizing the paper.

Author Response

(The authors gave the same response as above.)

Reviewer 3 Report

In this work, the authors have studied ML based algorithms for emotion interpretation. 

1. Please improve the image quality for fig 1. 2. Can you briefly explain the novelty of this work? You used a popular standard algorithm which is a bi-lstm for classification. I struggle to identify any novel contributions here.  3. The choice of baselines is not good. A bi-lstm with more parameters will surely outperform K-NNs and  SVMs. Can you add why you need a bi-lstm and not just an unidirectional LSTM or a GRU? The paper does not dive deep into why you used BI-LSTM. 4. Can you compute the mean and std dev among the folds?

Author Response

(The authors gave the same response as above.)

Reviewer 4 Report

The paper, “A Novel Approach for Emotion Recognition Based on EEG Signal Using Deep Learning”

The proposed approach is analyzed clearly. Nonetheless, the submission has several weak points, following points are not clear:

·       It would be clearer to sequence the steps described in Figure 7.

·       It would be advisable to do an ANOVA analysis of the results in Tables 3 and 2.

·       No details of the parameter values used in the comparison with the KNN, SVM and RF algorithms are given.

Author Response

Thank you for your valuable comments. We have sent the replies to your comments as attachments. Thank you for evaluating our article. Please see the attachment. 

Round 2

Reviewer 3 Report

I suggest the authors edit out the portion of the manuscript under application results and discussion "It has been observed that deep learning is more effective than some machine learning algorithms." It is neither a novel observation or a contribution from this work.

I urge the authors to add a line in the experiments highlighting that bi-LSTMS with more parameters are expected to outperform KNNs and SVMs. I don't find the results interesting and it is generally expected. A more fairer comparison would be to run uni-lstm and show that bi-lstm is better and the future context is necessary. I strongly urge the authors to highlights these points and add a "Limitations of the study" section to point this out.

Author Response

Reviewer Comment #1: I suggest the authors edit out the portion of the manuscript under application results and discussion "It has been observed that deep learning is more effective than some machine learning algorithms." It is neither a novel observation or a contribution from this work.

Author Comment #1: Thank you for your valuable comment. As a result of the classification we have made, we have made such a sentence because we saw that the deep learning method is more effective than machine learning. The results of the other deep learning classifier performed using this data set are given in Table 5. The results obtained in these studies are also more successful than machine learning algorithms. Therefore, we do not consider it appropriate to change this sentence. We also mentioned the novelties of the study in the article.

Reviewer Comment #2: I urge the authors to add a line in the experiments highlighting that bi-LSTMS with more parameters are expected to outperform KNNs and SVMs. I don't find the results interesting and it is generally expected. A more fairer comparison would be to run uni-lstm and show that bi-lstm is better and the future context is necessary.

Author Comment #2: Thank you for your valuable comment. There are multiple parameters in deep learning. Putting the results of each of the parameters here both reduces the clarity of the article and takes time. We already mentioned earlier that the best parameters are evaluated. In addition to these, there are no multi-parameter results in any study in the literature. We therefore refuse to make this revision. The deep learning model that the study focuses on is the BiLSTM deep learning model. Using Uni-LSTM will deviate from the purpose of the study. Therefore, we did not make this classification.

Reviewer Comment #3: I strongly urge the authors to highlights these points and add a "Limitations of the study" section to point this out.

Author Comment #3: Thank you for your valuable comment. Apart from the novelties and contributions mentioned above, the limitations and disadvantages of our study are also mentioned in detail. We are quoted again here. These are also included in the paper. The limitations of existing studies can be expressed as follows:

•The concept of emotion is an abstract structure. It varies from person to person. Therefore, it is not possible to reach a definite conclusion no matter what study is done.

•In current studies, visual or aural stimuli are generally used separately. Using them separately does not have a sufficient stimulating effect.

•In some other studies in the literature, visual and aural stimuli are used at the same time, and videos or music clips are usually shown for this purpose. However, these stimuli are not as effective as computer games.

•Traditional EEG devices are generally used in current studies. This device is both difficult and costly to use.

The disadvantages of the study can be listed as follows:

•    Due to the large sample length of the signals in the data set, we could not process the raw data. We had to do feature extraction. This may have caused some information to be lost. Processing the raw data can increase the performance of the deep learning algorithm.

•    The performances of the deep learning model and other machine learning models vary according to the selected feature extraction methods. More information can be obtained by using other feature extraction methods. This can positively affect the performance of the methods.

•    Although the number of data is small, the proposed method has been successful. However, this does not mean that the model is reliable. To determine the reliability of the model, different data sets should be used or the information obtained from the data set used should be reproduced.

At this point, when we look at the comments of the reviewer, we may feel that the limits of the study are not mentioned. However, in the study, as emphasized again above, all the elements such as the advantages, disadvantages, limits, and innovations of the study are given in detail. Because, as mentioned, we are talking about teamwork that brought the relevant dataset to the literature.